# Revisiting the Predictability of Performative, Social Events

**Juan Carlos Perdomo** [1]

## Abstract

Social predictions do not passively describe the future; they actively shape it. They inform actions and change individual expectations in ways that influence the likelihood of the predicted outcome. Given these dynamics, to what extent can social events be predicted? This question was discussed throughout the 20th century by authors like Merton, Morgenstern, Simon, and others who considered it a central issue in social science methodology. In this work, we provide a modern answer to this old problem. Using recent ideas from performative prediction and outcome indistinguishability, we establish that one can always efficiently predict social events accurately, regardless of how predictions influence data. While achievable, we also show that these predictions are often undesirable, highlighting the limitations of previous desiderata. We end with a discussion of various avenues forward.

## 1. Introduction

Social predictions do not passively describe the future; they actively shape it. They inform actions and change expectations in ways that influence the likelihood of predicted events. For instance, economic forecasts influence market prices, election predictions influence voter turnout, and climate forecasts shape policies that impact weather patterns.

This dynamic, where predictions are *performative* and shape data distributions, is pervasive throughout social prediction. It is also a feature, not a bug. The overarching goal of building systems that make predictions about people is to inform actions that drive positive changes in the world. For example, we predict health outcomes to cure disease and poverty to alleviate it. While desirable, this dynamic also introduces methodological challenges and casts doubt on

[1]Department of Computer Science, Harvard University, Boston MA, USA. Correspondence to: Juan Carlos Perdomo <jcperdomo@g.harvard.edu>.

*Proceedings of the 42nd International Conference on Machine Learning*, Vancouver, Canada. PMLR 267, 2025. Copyright 2025 by the author(s).

the predictability of social events. Motivated by these issues, we revisit the following question:

> *Is it possible to efficiently find a predictor that actively influences the likelihood of observed events, yet still produces valid predictions?*

During the 20th century, multiple researchers recognized that social predictions shape social patterns and posed this question in their work. To give a few examples, in his 1928 thesis, Oskar Morgenstern argued that accurate economic forecasts are generally impossible since public predictions can be self-negating (Morgenstern, 1928). Later, Simon (1954), as well as Grunberg & Modigliani (1954), used recently popularized fixed point theorems from topology to make progress on this problem and proved the *existence* of predictions that influence outcomes and are simultaneously calibrated. However, they did not address the algorithmic question of how one might actually *find* these predictors efficiently (both computationally and statistically speaking).

In this work, we revisit this question using modern mathematical tools. Framing our analysis in the language of *performative prediction* – a recent learning-theoretic framework introduced by Perdomo et al. (2020) that formalizes the causal aspects of social prediction – we establish the following result. One can always efficiently find predictions that actively shape the data distribution over outcomes and simultaneously satisfy rigorous validity guarantees like multi-calibration (Hébert-Johnson et al., 2018) or outcome indistinguishability (Dwork et al., 2021). Moreover, the statistical and computational complexity of finding these predictors is, in many cases, just as easy as in supervised learning, where the distribution is static.

However, while these rigorously calibrated predictions are always achievable, they are not always desirable. The fact that predictions are performative and influence the likelihood of future events invalidates traditional solution concepts inherited from supervised learning like calibration. Through simple examples, we show that these prediction rules can lead to poor equilibria where predictions exactly match the conditional distribution over outcomes – and hence are perfectly multicalibrated – while simultaneously explaining none of the variance in outcomes.

## 1.1. Overview of Technical Results

In this work, we study the problem of finding prediction rules $f$ that actively shape the distribution of observed outcomes $y$, yet still produce a valid forecast.

We cast this problem in the language of performative prediction (Perdomo et al., 2020), a recent learning theoretic framework that formalizes how data in social contexts is not static, but rather a function of the published predictor $f$. The key conceptual device of the framework is the notion of the *distribution map* $\mathcal{D}(\cdot)$, a function which maps predictors $f$ to distributions over feature, outcome pairs $(x, y)$. It captures how deploying different predictors induces different distributions.

In this work, we focus on the outcome performative case, in which predictions shape the distribution over outcomes $y$ but not features $x$. In particular, given a randomized predictor $f$, we write $(x, p, y) \sim \mathcal{D}(f)$ as shorthand for $x \sim \mathcal{D}_x, p \sim f(x)$, and $y \sim \mathcal{D}_y(x, p)$. The distribution $\mathcal{D}_x$ over $x$ is static, a prediction $p$ is sampled from $f(x)$, and the conditional distribution over outcomes $y$ is any function of $x$ and the forecaster's revealed prediction $p \in \mathbb{R}$. This setup exactly captures the prediction dynamics present in domains like health or education (see e.g. Perdomo et al. (2023) for a real world example). The learner makes a prediction $p$ regarding someone's individual outcome (future heart disease) using historical features (age, smoking history). The prediction influences treatments and individual behavior in ways that shape the likelihood of the predicted outcome (disease), but not the historical realization of features.

We assert that a forecast is valid if it is *performatively multicalibrated*, or synonymously, *indistinguishable*. Intuitively, a predictor is multicalibrated if $f(x)$ equals $y$ (in expectation), not just overall, but also when we condition on information about the individual $x$ and the prediction $p \sim f(x)$. A predictor is outcome indistinguishable if its correctness cannot be efficiently falsified, on the basis of the observed data and a pre-specified collection of computational tests. We use the terms multicalibration and (outcome) indistinguishability interchangeably since, interestingly, both conditions are formally equivalent (Dwork et al., 2021). We formally define these concepts for our performative setting below:

**Definition 1.1.** A randomized predictor $f$ mapping features $x$ to predictions $p$ is $\varepsilon$ performatively multicalibrated (indistinguishable) with respect to $\mathcal{C} \subseteq \{\mathcal{X} \times \mathbb{R} \to \mathbb{R}\}$ if

$$\left| \mathbb{E}_{\substack{x \sim \mathcal{D}_x, p \sim f(x) \\ y \sim \mathcal{D}_y(x,p)}}[c(x,p)(y-p)] \right| \leq \varepsilon \quad \text{for all } c \in \mathcal{C}.$$

Here, $\mathcal{C}$ parametrizes the degree to which predictions match outcomes. If $\mathcal{C}$ consists of just the constant 1 function, we recover the guarantee considered in Simon (1954) and Grunberg & Modigliani (1954) that $\mathbb{E}_{\mathcal{D}(f)}[y] = \mathbb{E}_{\mathcal{D}(f)}[f(x)]$

overall. If $\mathcal{C}$ consists of all bounded, measurable functions, then $f(x)$ must be equal to the conditional distribution over outcomes, $f(x) = \mathbb{E}_{\mathcal{D}(f)}[y|x]$ for all $x$. We can interpolate between these two extremes by varying $\mathcal{C}$.

While the guarantee in Definition 1.1 is known to be achievable in supervised learning where $\mathcal{D}(f) = \mathcal{D}(f')$ for all $f$ and $f'$, the situation is substantially more complicated in this performative setting. Predictions can, in general, be *self-negating*, and outcomes can move away from the published prediction $p \sim f(x)$. Given this possibility, it is not immediately obvious that performatively calibrated $f$ even exist (that is $f$ such that $\mathbb{E}_{\mathcal{D}(f)}[f(x)] = \mathbb{E}_{\mathcal{D}(f)}[y]$) without making strong regularity assumptions on $\mathcal{D}(\cdot)$. The situation is further complicated because the distribution map $\mathcal{D}(\cdot)$ is unknown to the learner. They can only deploy a predictor $f$ and observe the induced samples $(x, p, y) \sim \mathcal{D}(f)$.

Our first result shows that this guarantee Definition 1.1 is, in fact, efficiently achievable both statistically and computationally, thereby providing a modern learning-theoretic answer to the question posed by Morgenstern, Simon, and others. Unlike the bulk of work in performative prediction (Perdomo et al., 2020; Mendler-Dünner et al., 2020), this result requires no smoothness or continuity assumptions on the outcome performative distribution map $\mathcal{D}(\cdot)$ other than the condition that outcomes $y$ lie in a bounded range.

*Theorem* (Informal). Let $\mathcal{A}$ be any online algorithm which is guaranteed to produce predictions $p_t$ such that for any adversarially chosen sequence $\{(x_t, y_t)\}_{t=1}^T$,

$$\left| \sum_{i=t}^T c(x_t, p_t)(y_t - p_t) \right| \leq \mathsf{Regret}(T) \text{ for all } c \in \mathcal{C}. \quad (1)$$

Then, given $n$ draws from $\mathcal{D}(\cdot)$, the outputs of $\mathcal{A}$ can be converted into a randomized predictor $f_{\mathcal{A}}$ s.t. with probability $1 - \delta$ over the randomness of the draws from $\mathcal{D}(\cdot)$,

$$\left| \mathbb{E}_{(x,y,p) \sim \mathcal{D}(f)}[c(x,p)(y-p)] \right|$$
$$\leq \frac{\mathsf{Regret}(n)}{n} + \sqrt{\frac{\log(|\mathcal{C}|) + \log(1/\delta)}{n}} \quad \text{for all } c \in \mathcal{C}.$$

Furthermore, if $\mathcal{A}$ runs in $\mathsf{time}(t)$ at round $t$, then $f_{\mathcal{A}}$ can be computed in time $\mathcal{O}(n \cdot \mathsf{time}(n))$.[1]

The result establishes an efficient reduction from our main problem to an online multicalibration problem for which numerous algorithms now exist (Vovk, 2007; Foster & Kakade, 2006; Dwork et al., 2025; Gupta et al., 2022; Okoroafor et al., 2025). In particular, these algorithms can *efficiently*

---

[1]Recall that $(x, p, y) \sim \mathcal{D}(f)$ is shorthand for the sampling process $x \sim \mathcal{D}_x, p \sim f(x), y \sim \mathcal{D}_y(x, p)$. We abuse notation and write $(x, y) \sim \mathcal{D}(f)$ if the forecast $p$ does not appear in the expectation, but data $(x, y)$ is still that induced by $f$. For deterministic functions $h$, we write $(x, y) \sim \mathcal{D}(h)$ as shorthand for $x \sim \mathcal{D}_x, y \sim \mathcal{D}_y(x, h(x))$.

achieve the online guarantee in Equation (1) for various rich classes of functions $\mathcal{C}$ such as low-degree polynomials, decision trees, or, more generally, any $\mathcal{C}$ that is (weak agnostically) learnable or that belongs to a reproducing kernel Hilbert space. Moreover, these algorithms have $\sqrt{T}$ regret, implying that $f$ achieves the optimal $\mathcal{O}(n^{-1/2})$ rate for this problem. The reduction is conceptually simple, yields tight bounds on the generalization error, and requires no Lipschitzness conditions on the distribution map $\mathcal{D}(\cdot)$. That is, the conditional distribution over outcomes $\mathcal{D}_y(x, p)$ can be any discontinuous, non-smooth function of the forecast $p$. We only need to assume that outcomes are bounded.

Taking a step back, the theorem states that there are learning algorithms that can efficiently cope with social feedback. They make predictions that actively shape future events while still providing an honest, calibrated signal of the outcome. The fact that these prediction rules exist is fascinating and motivates us to ask further questions. Are these prediction rules truly desirable? What does it mean to make a good prediction if the prediction itself influences outcomes? Part of the answer to this question involves rethinking what the goals of prediction should be and how these goals are reflected in our choice of loss function. See (Kim & Perdomo, 2023) and (Miller et al., 2021) for further discussion.

However, leaving the choice of loss function aside and restricting ourselves to predictive accuracy as the main criterion, prediction rules satisfying the historical desiderata formalized in Definition 1.1 can be arbitrarily poor. We show that it is possible for a predictor $f$ to be performatively multicalibrated with respect to *all (continuous) functions* $c(x, p)$ while simultaneously explaining none of the variance in outcomes. We state the following result in terms of the core solution concepts from the performative prediction framework —performative stability and performative optimality — whose definitions we review below.[2]

*Theorem* (Informal). Assume that $y$ is binary and let $\mathcal{H} \subseteq \{\mathcal{X} \to [0, 1]\}$ be a benchmark class. Any predictor $f$ that is performatively multicalibrated with respect to $\mathcal{H}$ (satisfies Definition 1.1 with $\mathcal{C} = \mathcal{H}$) is also performatively stable with respect to $\mathcal{H}$ and hence satisfies:

$$\mathbb{E}_{(x,p,y)\sim\mathcal{D}(f)}(y - p)^2 \leq \min_{h\in\mathcal{H}} \mathbb{E}_{(x,y)\sim\mathcal{D}(f)}(y - h(x))^2. \tag{2}$$

---

[2]The solution concepts of performative stability and optimality are defined with respect to general loss functions $\ell(x, y; h)$ that can capture a wide variety of higher level objectives. For instance, we can choose to have losses which encourage forecasts to match outcomes, $\ell(x, y; h) = (h(x) - y)^2$, or losses which encourage the induced distributions $\mathcal{D}(f)$ towards a particular outcome $\ell(x, y; h) = -y$. See (Kim & Perdomo, 2023). Since the goal here is to understand the relationship between from Definition 1.1 and predictive accuracy in performative contexts, we specialize the definitions to the case of squared loss for simplicity.

However, there exist a distribution map $\mathcal{D}(\cdot)$ such that $f$ can be performatively multicalibrated with respect to all bounded functions $c(x, p)$, yet simultaneously maximize the performative risk for any $\mathcal{H}$,

$$\mathbb{E}_{(x,p,y)\sim\mathcal{D}(f)}(y - f(x))^2 \geq \max_{h\in\mathcal{H}} \mathbb{E}_{(x,y)\sim\mathcal{D}(h)}(y - h(x))^2. \tag{3}$$

The theorem states that any predictor that is performatively multicalibrated with respect to $\mathcal{H}$ is also performatively stable with respect to $\mathcal{H}$. A predictor $f$ is performatively stable if it induces a distribution $\mathcal{D}(f)$ such that no model $h \in \mathcal{H}$ has a lower loss over $\mathcal{D}(f)$. This is Equation (2). If we judge the performance of a performatively stable predictor $f$ (and the alternatives $h$) purely based on the distribution $\mathcal{D}(f)$, there is no reason to switch to an alternative model in $\mathcal{H}$ since these have higher loss over $\mathcal{D}(f)$.

However, stability ignores the fact that different predictors induce different distributions. Note that the expectation in the right hand side of the stability condition, Equation (2), is taken over $\mathcal{D}(f)$ not $\mathcal{D}(h)$. Yet, the true measure of performance for a predictor $h$ in performative contexts is its *performative risk* – the expected loss over its own induced distribution – formally defined as $\mathbb{E}_{\mathcal{D}(h)}(y - h(x))^2$. Note that the predictor inside the loss function and the predictor in the distribution map $\mathcal{D}(\cdot)$ are now the same. A model $f$ is performatively optimal with respect to a class $\mathcal{H}$ if it has lower performative risk than any model in $h \in \mathcal{H}$. Equation (3) states that it is possible for a predictor to be performatively multicalibrated with respect to all functions $c(x, p)$ while simultaneously *maximizing* the performative risk.[3]

Note that our intuition from supervised learning is exactly reversed. In supervised learning, where $(x, y) \sim \mathcal{D}$, if $f(x) = \mathbb{E}[y \mid x]$ for all $x$, then $\mathbb{E}_{\mathcal{D}}(y - f(x))^2 \leq \mathbb{E}_{\mathcal{D}}(y - h(x))^2$ for any function $h$. In performative prediction, where $f$ influences the distribution over $(x, y)$, the result shows it is possible for $f(x) = \mathbb{E}_{\mathcal{D}(f)}[y \mid x]$ for every $x$, yet $\mathbb{E}_{\mathcal{D}(f)}(y - f(x))^2 \geq \mathbb{E}_{\mathcal{D}(h)}(y - h(x))^2$. The inequality is now flipped because of performativity.

As a whole, our results advance the mechanics and algorithmic foundations of social prediction, explaining how and why one can find predictors that dynamically shape yet also rigorously predict social outcomes. In doing so, we shed new light on old questions previously considered by leading researchers of the 20th century. Lastly, by re-evaluating historical desiderata using a modern lens, we highlight ways in which previous debates fall short and inspire new debate regarding what the broader goals of prediction in the social world should be.

---

[3]There is a non-trivial gap for this problem, $\min_h \mathbb{E}_{\mathcal{D}(h)}(y - h(x))^2 \ll \max_h \mathbb{E}_{\mathcal{D}(h)}(y - h(x))^2$. See Section 5.

## 1.2. Related Work

**Social Science.** As discussed in the introduction, several authors across the social sciences have studied the question at the heart of our work regarding the predictability of social events. In addition to Morgenstern (1928), Simon (1954) states this exact problem in the introduction to his paper, where he remarks that he learned about the issue from Hayek (1944). The existence, but not computation, of a predictor solving this problem is also considered in Grunberg & Modigliani (1954). Their paper inspired our choice of title.

Our problem also lies at the heart of the famous Lucas critique (Lucas, 1976), which marked a turning point in macroeconomic theory. Outside of economics, this problem has been extensively studied in sociology dating back to work by Merton (1948) and more recently by MacKenzie (2008). In philosophy, it is studied in Buck (1963). Our results complement prior work by advancing our algorithmic understanding of the problem and establishing how these prediction problems can be efficiently solved both statistically and computationally.

**Performative Prediction.** The area of performative prediction was initiated by Perdomo et al. (2020), who proposed a formal framework to study predictions that shape data distributions. We cannot cover all the work in this growing field, but we point the reader to the excellent survey by Hardt & Mendler-Dünner (2025) for a broader overview. Within this literature, our results are most closely related to the literature on finding performatively stable points (Mendler-Dünner et al., 2020; Drusvyatskiy & Xiao, 2023; Mofakhami et al., 2023; Oesterheld et al., 2023; Mofakhami, 2024; Taori & Hashimoto, 2023; Khorsandi et al., 2024). Relative to these analyses, our results differ since we make no smoothness assumptions on the way forecasts influence outcomes and restrict ourselves to the outcome performative setting where predictions only influence the distribution over the outcomes, but not features.

**Multicalibration & Outcome Indistinguishability.** Our results build on the recent lines of work on multicalibration (Hébert-Johnson et al., 2018) and outcome indistinguishability (Dwork et al., 2021), further extending the foundations of these ideas to non-supervised learning settings. Our proofs heavily rely on new algorithmic insights from online calibration, pioneered by Foster & Vohra (1998), and extended by (Foster & Kakade, 2006; Kakade & Foster, 2008; Vovk, 2007; Gupta et al., 2022) and others. Within this broad literature, our work is most closely related to Kim & Perdomo (2023), who used these tools developed in Gopalan et al. (2023) to compute performatively optimal (not stable) predictors for a restricted version of the outcome performativity setting. In their setup, outcomes are binary and their conditional distribution is a function of $x$ and a discrete decision

$\hat{y}$ belonging to a finite set. Our setup generalizes theirs since both outcomes $y$ and forecasts $p$ can be real-valued.

## 2. Preliminaries

Before presenting our main results, we review some technical preliminaries and provide some background and motivation behind our core solution concepts.

**The Distribution Map.** We assume throughout that data $(x, p, y) \sim \mathcal{D}(f)$ is generated according to the following process. The learner publishes a predictor $f : \mathcal{X} \to \Delta([0,1])$ mapping features $x$ to a distribution $f(x)$ over the unit interval $[0,1]$. Features $x$ are drawn i.i.d from a fixed distribution $\mathcal{D}_x$ over an arbitrary set $\mathcal{X}$, predictions $p \in [0,1]$ are sampled from $f$, $p \sim f(x)$. Outcomes $y$ are sampled from $\mathcal{D}_y(x, p)$ where $\mathcal{D}_y(x, p)$ is any distribution over the unit interval $[0,1]$. Our results apply to any $\mathcal{D}_y(x, p)$ supported on a bounded interval by rescaling.

**Performative Multicalibration.** Calibration is the *sine qua non* definition of validity for a probabilistic forecast (Dawid, 1985). For binary $y$, a predictor is calibrated if conditional on $f(x) = v$ for $v \in [0, 1]$, the outcome $y$ occurs a $v$ fraction of the time, $\Pr[y = 1 \mid f(x) = v] = v$. Multicalibration (Hébert-Johnson et al., 2018) is a strengthening of calibration, requiring that $f(x) = y$, not just overall, but also once we condition on $x$ belonging to any set $G$ in a collection $\mathcal{G}$: $\Pr[y = 1 \mid f(x) = v, x \in G] = v$ for all $G \in \mathcal{G}$. This condition is equivalent, up to a normalization factor of $\Pr[f(x) = v, x \in G]$, to the one we wrote in Definition 1.1 since letting $c_{G,v}(x, p) = 1\{x \in G, p = v\}$, we can write:

$$\mathbb{E}_{(x,p,y)\sim\mathcal{D}(f), p\sim f(x)}[c_{G,v}(x,p)(y-p)] = $$
$$(\Pr[y = 1 \mid f(x) = p, x \in G] - v)\Pr[f(x) = v, x \in G].$$

Conversely, a predictor $f$ is outcome indistinguishable (or OI) (Dwork et al., 2021) if it establishes a generative model that cannot be falsified on the basis of a pre-specified collection of tests or distinguishers $A \in \mathcal{F} \subseteq \{\mathcal{X} \times [0, 1] \times [0, 1] \to \mathbb{R}\}$. Simplifying our discussion to the case of a binary outcome, each distinguisher takes features $x$, predictions $p$, and an outcome $y$ and outputs 1 or 0 (i.e. is this a real outcome/prediction for $x$). A predictor is OI if all the distinguishers in the collection $\mathcal{F}$ behave the same when given the true outcome $y \sim \mathcal{D}(f)$ versus an outcome sampled from the model $\tilde{y} \sim \text{Ber}(p), p \sim f(x)$. Extended to the performative context, $f$ is outcome indistinguishable with respect to $\mathcal{F}$ if for all $A \in \mathcal{F}$

$$\mathbb{E}_{(x,p,y)\sim\mathcal{D}(f)}A(x,p,y) = \mathbb{E}_{x\sim\mathcal{D}_x p\sim f(x), \tilde{y}\sim\text{Ber}(p)}A(x,p,\tilde{y}) \tag{4}$$

For binary $y$, this guarantee is, in fact, equivalent to our performative multicalibration guarantee from Definition 1.1 since this above equation is true if and only

if $\mathbb{E}_{(x,p,y)\sim\mathcal{D}(f)}[c(x,p)(y - p)] = 0$ for $c_A(x,p) = A(x,p,1) - A(x,p,0)$, as seen in Dwork et al. (2021) for the supervised learning case.

We feel that this rewriting is particularly insightful. Note that outcomes on the left of Equation (4) are performative: They are sampled from $\mathcal{D}(f)$. However, outcomes on the right hand side are sampled according to $f$. The predictor $f$ is actively influencing the distribution in such a way that outcomes behave as if they were sampled according to its own predicted distribution. By observing samples $(x,y) \sim \mathcal{D}(f)$, we might seemingly believe that $f$ is not influencing the data at all! It's just a great predictor.

However, this intuition is false. A predictor $f$ can pass *all* bounded tests $\mathcal{F}$ and be multicalibrated with respect to any collection $\mathcal{C}$ while simultaneously explaining none of the variance in $y$. We present this construction in Section 5.

## 3. Algorithmic Results

This section presents our core algorithmic results, illustrating how one can find prediction rules that influence data and are indistinguishable from the true outcomes. These algorithmic procedures are conceptually simple, computationally efficient, and near statistically optimal.

**Technical Overview.** Recall that the goal is to find a prediction rule $f$ satisfying the following indistinguishability guarantee from Definition 1.1. For all $c \in \mathcal{C}$,

$$\left|\mathbb{E}_{(x,p,y)\sim\mathcal{D}(f)}[(y - p)c(x,p)]\right| \leq \varepsilon. \tag{5}$$

Rather than solving this problem directly, we reduce it to a seemingly harder, *online* problem and then perform an online-to-batch conversion. That is, we show that any online algorithm $\mathcal{A}$ that (deterministically) produces predictions $p_t = f_t(x_t)$ such that $|\sum_{t=1}^{T} c(x_t, p_t)(y_t - p_t)| \leq o(T)$, for any adversarial sequence of $\{(x_t, y_t)\}_{t=1}^{T}$ can be converted to a batch predictor $f$ satisfying the indistiguishability guarantee from Equation (5). We begin by formally defining the online protocol used in our reduction.

**Definition 3.1** (Online Prediction). Online prediction is a sequential, two-player game between a Learner and Nature. At every round $t$, the Learner moves first and selects a function $f_t : \mathcal{X} \to [0, 1]$, deterministically mapping $x \in \mathcal{X}$ to predictions $p_t$ in $[0, 1]$, as a function of the history up until time $t$. Nature moves second and selects $(x_t, y_t) \in \mathcal{X} \times \mathcal{Y}$ with knowledge of $x_t$ and $p_t = f_t(x_t)$. We refer to the sequence $\{(x_t, y_t, f_t, p_t)\}_{t=1}^{T}$ as the transcript of the game.

Online prediction is a classical problem in statistics, game theory, and machine learning for which numerous algorithms have been developed (Cesa-Bianchi & Lugosi, 2006; Foster & Vohra, 1998; Kakade & Foster, 2008). Our definition differs slightly from traditional presentations since

we assume that the learner moves first and commits to a function $f_t$ at every round before seeing $x_t$ (instead of moving second and producing a prediction $p_t$ after seeing the features $x_t$). However, this is just a difference in style, not substance. Most online algorithms in the literature also commit to a prediction rule $f_t : \mathcal{X} \to [0, 1]$ at every round before seeing the features $x_t$. Because $f_t$ is revealed, Nature knows $p_t = f_t(x_t)$ for every $x_t$ and can use this information (potentially adversarially) when choosing $x_t$ and $y_t$. Moving on, our results rely on algorithms that achieve the following guarantee in online prediction.

**Definition 3.2** (Online Multicalibration). Let $\mathcal{C} \subseteq \{\mathcal{X} \times [0, 1] \to \mathbb{R}\}$ be a class of functions. An algorithm $\mathcal{A}$ guarantees online multicalibration with respect to a set $\mathcal{C}$ at a rate bounded by $\mathsf{Regret}_{\mathcal{A}}(\cdot)$ if it always generates a sequence of functions $f_t$ for the Learner that, no matter Nature's strategy in the online protocol (Definition 3.1), will yield a transcript $\{(x_t, y_t, f_t, p_t)\}_{t=1}^{T}$ satisfying,

$$\left|\sum_{t=1}^{T} c(x_t, p_t)(p_t - y_t)\right| \leq \mathsf{Regret}_{\mathcal{A}}(T),$$

for all $c \in \mathcal{C}$ where $\mathsf{Regret}_{\mathcal{A}}(T) : \mathbb{N} \to \mathbb{R}_{\geq 0}$ is $o(T)$.

Algorithms that achieve this guarantee date back to the work Vovk (2007); Vovk et al. (2005a) and Foster & Kakade (2006) (they refer to it under different names like *resolution* or just *calibration*). Following the work of Hébert-Johnson et al. (2018) introducing multicalibration, there has been a flurry of recent papers introducing new algorithms for this problem (Dwork et al., 2025; Gupta et al., 2022; Garg et al., 2024; Okoroafor et al., 2025). We will make use of these procedures when instantiating our general results. As a final preliminary step, we define the online-to-batch procedure we use in our analysis:

**Definition 3.3** (Batch Converstion). Fix an outcome performative distribution map $\mathcal{D}(\cdot)$. Let $\{(x_t, y_t, f_t, p_t)\}_{t=1}^{T}$ be the transcript generated in the online prediction protocol (Definition 3.1) where at every time step $t$, the Learner chooses $f_t$ according to $\mathcal{A}$, and Nature selects features $x_t$ and outcomes $y_t$, by sampling them from the distribution map: $x_t \sim \mathcal{D}_x, y_t \sim \mathcal{D}(x_t, p_t)$ where $p_t = f_t(x_t)$.

Define the $T$-round batch version of $\mathcal{A}$, $f_{\mathcal{A}} : \mathcal{X} \to \Delta([0, 1])$ to be the randomized predictor that given $x$, selects $f_i \in \{f_1, \ldots, f_T\}$ from the transcript uniformly at random, and then predicts $p = f_i(x)$.

This style of online-to-batch conversion, where one uniformly randomizes over all previous predictors, is standard in the online learning literature and is the typical starting point when converting online algorithms to batch learners (see, e.g., Gupta et al. (2022); Okoroafor et al. (2025)). The main difference in this construction is that samples $(x_t, y_t)$

are not drawn i.i.d from a fixed distribution $\mathcal{D}$, but rather from the distribution $(x_t, y_t) \sim \mathcal{D}(f_t)$ induced by the predictions. Note that while each of the individual functions $f_t \in \{f_1, \ldots, f_t\}$ in the transcript are deterministic, the batch predictor is *randomized* since it first mixes over the choice of $f_t$. With these definitions out of the way, we can now state the main theorem for this section:

**Theorem 3.4.** *Let $\mathcal{C} \subseteq \{\mathcal{X} \times [0,1] \to [-1,1]\}$ be a class of functions and let $\mathcal{A}$ be an algorithm that guarantees online multicalibration with respect to $\mathcal{C}$ at a rate bounded by* $\mathsf{Regret}_{\mathcal{A}}(\cdot)$ *(Definition 3.2). Define $f_{\mathcal{A}}$ to be the batch predictor of $\mathcal{A}$ trained on $n$ rounds of interaction (see Definition 3.3). Then, with probability $1 - \delta$ over the samples drawn from $\mathcal{D}(\cdot)$, for all $c \in \mathcal{C}$,*

$$\left| \mathbb{E}_{(x,p,y) \sim \mathcal{D}(f_{\mathcal{A}})} c(x,p)(p - y) \right|$$
$$\leq \frac{\mathsf{Regret}_{\mathcal{A}}(n)}{n} + 4\sqrt{\frac{\log(|\mathcal{C}|) + \log(1/\delta)}{n}}.$$

*Proof Sketch.* The proof follows the standard template for online-to-batch conversions. Since outcomes are a function of $x$ and specific prediction $p$ we can use the definition of $f_{\mathcal{A}}$ to decompose the left hand side as:

$$\mathbb{E}_{\substack{x \sim \mathcal{D}_x, p \sim f_{\mathcal{A}}(x) \\ y \sim \mathcal{D}_y(x,p)}} c(x,p)(p - y) =$$
$$\frac{1}{n} \sum_{i=1}^{n} \mathbb{E}_{\substack{x \sim \mathcal{D}_x \\ y \sim \mathcal{D}_y(x, f_i(x_i))}} [c(x,p)(p - y) \mid f_{\mathcal{A}} = f_i],$$

Then, viewing the transcript $\{(x_t, y_t, f_t, p_t)\}_{t=1}^{T}$ as a stochastic process, we use a Martingale argument and apply the Azuma-Hoeffding inequality to establish a high-probability upper bound on the RHS. $\square$

As we mentioned, Theorem 3.4 is a general reduction. It states that any algorithm that is online multicalibrated with respect to a class of functions $\mathcal{C}$ yields a (batch) prediction rule $f_{\mathcal{A}}$ that is *performatively* multicalibrated with respect to $\mathcal{C}$. The reduction, moreover, is statistically efficient: the error decreases at the optimal $\mathcal{O}(n^{-1/2})$ rate and the dependence on the failure probability $\delta$ is logarithmic.[4] The $\log(|\mathcal{C}|)$ dependence comes from a standard union bound argument and can be sharpened using well-known techniques.[5] The reduction is also *computationally* efficient. If the runtime of the online algorithm $\mathcal{A}$ is bounded by $\mathsf{time}(t)$

---

[4]The dependence on $n$ cannot be improved without further assumptions. In particular, take the case where $\mathcal{C}$ only contains the constant one function, $y$ is binary, and $\mathcal{D}(\cdot)$ is not performative so that $(x,y) \sim \mathcal{D}_*$ where $\mathcal{D}_*$ is a fixed distribution. In this case, the problem becomes mean estimation for a Bernoulli random variable, which has a well-known $\Omega(n^{-1/2})$ lower bound (see e.g. Anthony & Bartlett (2009))

[5]One can, under further assumptions, replace $\log(|\mathcal{C}|)$ with a norm-based bound (Cesa-Bianchi et al., 2004).

at time step $t$, then the online-to-batch conversion takes time $\mathcal{O}(n \cdot \mathsf{time}(n))$. Therefore, if $\mathcal{A}$ runs in polynomial time, so does the batch version $f_{\mathcal{A}}$.

**Example Instantiations.** Having introduced the main result, we now illustrate how it can be instantiated using existing algorithms to guarantee multicalibration with respect to rich classes of functions $\mathcal{C}$. Since there are by now many different online algorithms that satisfy Definition 3.2, these examples are by no means meant to be exhaustive. We simply wish to illustrate some interesting cases with the understanding that there are numerous alternatives. Our end-to-end results can be improved as the community develops online algorithms with sharper regret bounds or better runtimes that can be plugged into Theorem 3.4.

In particular, we instantiate our general reduction using the K29 algorithm from (Vovk et al., 2005a;b). This algorithm is a simple, kernel-based procedure which can efficiently guarantee calibration with respect to functions in a Reproducing Kernel Hilbert Space. We present a self-contained description and analysis of the algorithm in Appendix A. The following corollary summarizes some of its implications.

**Corollary 3.5.** *The following statements are true.*

*(a) Any Finite Collection. Let $\mathcal{C} \subseteq \{\mathcal{X} \times [0,1] \to [-1,1]\}$ be any finite set of functions $c(x,p)$ that are continuous in the forecast $p$. Then, there exists a choice of kernel, such that the K29 algorithm is online multicalibrated with respect to $\mathcal{C}$ at a rate bounded by $\sqrt{T|\mathcal{C}|}$. Consequently, the batch version $f$ trained on $n$ rounds of interaction satisfies:*

$$\left| \mathbb{E}_{(x,y) \sim \mathcal{D}(f), p \sim f(x)} c(x,p)(p - y) \right| \leq$$
$$\sqrt{\frac{|\mathcal{C}|}{n}} + 4\sqrt{\frac{\log(|\mathcal{C}|) + \log(1/\delta)}{n}} \quad \text{for all } c \in \mathcal{C}.$$

*Furthermore, the per-round run-time is $\widetilde{\mathcal{O}}(t \cdot |\mathcal{C}|)$. Hence, $f$ runs in time $\mathcal{O}(n^2 |\mathcal{C}|)$.*

*(b) Linear Functions. Define $\mathcal{X} = \{x \in \mathbb{R}^d : \|x\|_2 \leq 1\}$, $\mathcal{C}_{\mathrm{lin}} = \{\theta^\top x + p : \|\theta\|_2 \leq 1\}$, and let $\mathcal{C}$ be a finite subset of $\mathcal{C}_{\mathrm{lin}}$. Then, there exists a choice of kernel such that the K29 algorithm is online multicalibrated with respect to $\mathcal{C}_{\mathrm{lin}}$ at a rate bounded by $\sqrt{2T}$. Consequently, the batch version $f$ of this procedure trained on $n$ rounds of interaction satisfies:*

$$\left| \mathbb{E}_{(x,y) \sim \mathcal{D}(f), p \sim f(x)} c(x,p)(p - y) \right| \leq$$
$$\sqrt{\frac{2}{n}} + 4\sqrt{\frac{\log(|\mathcal{C}|) + \log(1/\delta)}{n}} \quad \text{for all } c \in \mathcal{C}.$$

*Furthermore, the per-round run-time is bounded by $\widetilde{\mathcal{O}}(d)$. Hence, $f$ runs in time $\widetilde{\mathcal{O}}(n \cdot d)$.*

*(c) Low-Degree Boolean Functions. Define $\mathcal{X} = \{0,1\}^d$, and $\mathcal{C}_{\mathrm{LowDeg}} \subseteq \{\mathcal{X} \to [-1,1]\}$ to be the set of Boolean*

*functions of degree s. That is, those that can be written as,*

$$c(x) = \sum_{S \subseteq [d], |S| \le s} \alpha_S \prod_{i \in S} x_i$$

*for some coefficients $\{\alpha_S\}_{S \subset [n]} \in \mathbb{R}$. Let $\mathcal{C}$ be any finite subset of $\mathcal{C}_{\mathrm{LowDeg}} \cup \{c(x,p) = p\}$. Then, there exists a choice of kernel such that the K29 algorithm is online multi-calibrated with respect to $\mathcal{C}_{\mathrm{LowDeg}} \cup \{c(x,p) = p\}$ at rate bounded by $10\sqrt{d^s \cdot T}$.*

*Consequently, the batch version $f$ of this procedure trained on $n$ rounds of interaction satisfies:*

$$\left| \mathbb{E}_{(x,y) \sim \mathcal{D}(f), p \sim f(x)} c(x,p)(p-y) \right| \le$$
$$10\sqrt{\frac{d^s}{n}} + 4\sqrt{\frac{\log(|\mathcal{C}|) + \log(1/\delta)}{n}} \quad \textit{for all } c \in \mathcal{C}$$

*Lastly, the per-round run-time of the algorithm is $\widetilde{\mathcal{O}}(t \cdot ds)$. Hence, $f$ runs in time $\mathcal{O}(ds \cdot n^2)$.*

The corollary illustrates how one can efficiently find prediction rules $f$ that are performatively multicalibrated with respect to common classes of functions, for instance, linear functions or low-degree polynomials. One can even efficiently guarantee indistinguishability with respect to any polynomially sized collection $\mathcal{C}$ (as long as the functions $c \in \mathcal{C}$ are each efficiently computable). This, in particular, implies performative indistinguishability with respect to rich classes of functions, like deep neural networks. These hold without making any Lipschitzness assumptions on the distribution map $\mathcal{D}(\cdot)$. Furthermore, the runtime of the procedures is, up to small polynomial factors, no different than that of algorithms achieving solving the analogous guarantee in supervised learning settings.

## 4. Structural Results

In the previous section, we illustrated how one can find predictors $f$ that actively shape the data $(x,y) \sim \mathcal{D}(f)$, yet still make predictions that are computationally indistinguishable from the true outcomes. Here, we take a step further and ask: are these computationally indistinguishable predictors useful (in a risk minimization sense)? What is the relationship between calibration and loss minimization in this outcome performative context?

To answer these, we analyze the relationship between the previous historical desiderata (Definition 1.1) and the core solutions in performative prediction. We review these below.

**Definition 4.1** (Performative Stability and Optimality). Let $\ell$ be a loss function an $\mathcal{H} \subseteq \{\mathcal{X} \to [0,1]\}$ a benchmark class. A predictor $f_{\mathrm{ps}}$ is performatively stable with respect to the class $\mathcal{H}$ if,

$$\mathbb{E}_{(x,p,y) \sim \mathcal{D}(f_{\mathrm{ps}})} \ell(p,y) \le \min_{h \in \mathcal{H}} \mathbb{E}_{(x,y) \sim \mathcal{D}(f_{\mathrm{ps}})} \ell(h(x), y).$$

On the other hand, a predictor $f_{\mathrm{po}}$ is performatively optimal with respect the class $\mathcal{H}$ if,

$$\mathbb{E}_{(x,p,y) \sim \mathcal{D}(f_{\mathrm{po}})} \ell(p,y) \le \min_{h \in \mathcal{H}} \mathbb{E}_{(x,y) \sim \mathcal{D}(h)} \ell(h(x), y).$$

Lastly, we refer to $\mathbb{E}_{(x,y) \sim \mathcal{D}(h)} \ell(h(x), y)$ as the performative risk of a predictor $h$.

Intuitively, a predictor $f$ is performatively stable if its optimality cannot be refuted on the basis of the data that it induces. If we evaluate the loss of any alternative predictor $h$ over the distribution $\mathcal{D}(f)$, we find it will have a higher loss than $f$ itself. However, this ignores the fact that different models induce distributions. Performatively optimality embraces this observation. A model $f$ is performatively optimal if it minimizes the performative risk.

As stated, our definitions are slight generalizations of those initially proposed by Perdomo et al. (2020) since we allow $f$ to be randomized and do not require that it be a member of the class $\mathcal{H}$. Furthermore, our definition holds for any possibly non-parametric class $\mathcal{H}$. However, if $\mathcal{H} = \mathcal{F}_\Theta$ is a parametric class and we impose that $f_{\mathrm{ps}} \in \mathcal{F}_\Theta$, we recover the previous definition,

$$\theta_{\mathrm{ps}} \in \arg\min_{\theta \in \Theta} \mathbb{E}_{(x,y) \sim \mathcal{D}(\theta_{\mathrm{ps}})} \ell(f_\theta(x), y) \iff$$
$$\mathbb{E}_{(x,y) \sim \mathcal{D}(\theta_{\mathrm{ps}})} \ell(f_{\theta_{\mathrm{ps}}}(x), y) \le \min_{\theta \in \Theta} \mathbb{E}_{(x,y) \sim \mathcal{D}(\theta_{\mathrm{ps}})} \ell(f_\theta(x), y).$$

The following theorem is the main result of this section, showing how performatively multicalibrated predictors as per Definition 1.1 are also performatively stable with respect to standard losses like squared error. To simplify our presentation, we make the further assumption that $y$ is binary.

**Theorem 4.2.** *Assume that outcomes $y$ are binary. Let $\mathcal{H} : \mathcal{X} \to [0,1]$ be a benchmark class and let $\ell$ be the squared loss, $\ell(p,y) = \frac{1}{2}(p-y)^2$. If $f$ is $\varepsilon$-performatively multicalibrated with respect to the functions $\mathcal{C}$, defined as, $\mathcal{C} = \{c(x,p) = p - 1/2\} \cup \{c(x,p) = h(x) - 1/2 : h \in \mathcal{H}\}$, then $f$ is $2\varepsilon$-performatively stable with respect to $\mathcal{H}$.*

The proof of this result follows from two lemmas. These make use of recent tools pioneered by Gopalan et al. (2023) for the supervised learning setting, showing how outcome indistinguishable predictors are loss-minimizing.

**Lemma 4.3.** *Assume that outcomes $y$ are binary and let $\ell : [0,1] \times \{0,1\} \to \mathbb{R}$ be a proper scoring rule. Fix a parameter $\varepsilon > 0$. If a function $f : \mathcal{X} \to [0,1]$ satisfies the following inequalities for all $h \in \mathcal{H}$,*

$$\mathbb{E}_{(x,p,y) \sim \mathcal{D}(f)} \ell(p,y) \le \mathbb{E}_{\substack{x \sim \mathcal{D}_x, p \sim f(x) \\ y \sim \mathrm{Ber}(p)}} \ell(p,y) + \varepsilon \quad (6)$$

$$\mathbb{E}_{\substack{x \sim \mathcal{D}_x, p \sim f(x) \\ y \sim \mathrm{Ber}(p)}} \ell(h(x),y) \le \mathbb{E}_{(x,y) \sim \mathcal{D}(f)} \ell(h(x),y) + \varepsilon$$

*then, $f$ is $2\varepsilon$-performatively stable with respect to $\mathcal{H}$.*

*Proof.* The proof follows immediately from the indistinguishability conditions and the assumption that $\ell$ is a proper loss. In particular, from the first condition we have that:

$$\mathbb{E}_{(x,p,y)\sim\mathcal{D}(f)}\ell(p,y) \leq \mathbb{E}_{x\sim\mathcal{D}_x,p\sim f(x),y\sim\text{Ber}(p)}\ell(p,y) + \varepsilon.$$

Next, the definition of a proper loss is that if $y \sim \text{Ber}(p)$, every other $h : \mathcal{X} \to [0,1]$ must have at least as high a loss,

$$\mathbb{E}_{x\sim\mathcal{D}_x,p\sim f(x),y\sim\text{Ber}(p)}\ell(p,y) \leq$$
$$\min_{h\in\mathcal{H}} \mathbb{E}_{x\sim\mathcal{D}_x,p\sim f(x),y\sim\text{Ber}(p)}\ell(h(x),y).$$

The result then follows from using the second assumption which guarantees that for any $h \in \mathcal{H}$,

$$\mathbb{E}_{x\sim\mathcal{D}_x,p\sim f(x),y\sim\text{Ber}(p)}\ell(h(x),y) \leq \qquad (7)$$
$$\mathbb{E}_{(x,y)\sim\mathcal{D}(f)}\ell(h(x),y) + \varepsilon.$$

Chaining all inequalities together, we get the $2\epsilon$ bound. $\square$

The inequalities in the assumptions of the lemma are best understood as a particular kind of loss outcome indistinguishability conditions (Gopalan et al., 2023). Recalling our discussion from Section 2, one can think of the losses $\ell(h(x),y)$ as a type of distinguisher $A_\ell(x,h(x))$. The lemma shows that if $f$ is a generative model of outcomes that passes a class of tests defined by $\ell$ and $\mathcal{H}$, then $f$ is performatively stable.

A similar condition had also been considered in the performative prediction literature by Kim & Perdomo (2023). The key conceptual difference is that (Kim & Perdomo, 2023) consider learning a model $\widetilde{\mathcal{D}}_y(x,p)$ that is indistinguishable from the true distribution map $\mathcal{D}_y(x,p)$ from the perspective of a set of tests $A_{\ell,h}$ (that depend on the set of benchmark functions $h$ and loss function $\ell$). Informally, $\widetilde{\mathcal{D}}_y(x,h(x)) \approx_{A_{\ell,h}} \mathcal{D}_y(x,h(x))$ for all $(h,\ell)$ in some set.

However, the indistinguishability conditions in Lemma 4.3 are with respect to a predictor $f : \mathcal{X} \to \Delta([0,1])$, not a model of the distribution map $\mathcal{D}_y : \mathcal{X} \times [0,1] \to \Delta([0,1])$. They relate the performative risk to the expected risk where outcomes are self-confirming and sampled from the model proposed by $f$. Note that $p \sim f(x)$ and $y \sim \text{Ber}(p)$ on the LHS of Equation (7). Stated again informally, these conditions require that $\widetilde{\mathcal{D}}_y(x,f(x)) \approx_{A_{\ell,h}} f(x)$. Whereas the indistinguishability criteria from (Kim & Perdomo, 2023) yield performative *optimality*, ours yield performative *stability*. Hence, the takeaway message from this discussion is that stability (not optimality) is the natural limit of deploying online calibration algorithms in performative contexts.

The next result finishes the proof of Theorem 4.2. It establishes that the loss OI condition from Lemma 4.3 is equivalent to performative multicalibration if we fix $\ell$ to be the squared loss.

**Lemma 4.4.** *If $\ell$ is the squared loss $\ell(p,y) = \frac{1}{2}(y-p)^2$, then*

$$\left|\mathbb{E}_{(x,p,y)\sim\mathcal{D}(f)}\ell(p,y) - \mathbb{E}_{\substack{x\sim\mathcal{D}_x,p\sim f(x)\\y\sim\text{Ber}(p)}}\ell(p,y)\right| \leq \varepsilon$$

$$\left|\mathbb{E}_{\substack{x\sim\mathcal{D}_x,p\sim f(x)\\y\sim\text{Ber}(p)}}\ell(h(x),y) - \mathbb{E}_{(x,y)\sim\mathcal{D}(f)}\ell(h(x),y)\right| \leq \varepsilon$$

*for all $h \in \mathcal{H}$ if and only if $f$ is $\varepsilon$-performatively multicalibrated with respect to the functions, $\mathcal{C} = \{c(x,p) = p - 1/2\} \cup \{c(x,p) = h(x) - 1/2 : h \in \mathcal{H}\}$.*

Theorem 4.2 follows directly from the last two lemmas. Moreover, it also follows from Corollary 3.5 that one can statistically and computationally efficiently find predictors $f$ that are performatively stable for important classes of benchmark classes $\mathcal{H}$ by via an online-to-batch conversion.

These results strengthen our previous understanding of performative stability. In the initial work on performative prediction, stable predictors were only known to be efficiently computable under strong regularity conditions on the distribution map $\mathcal{D}(\cdot)$ (Perdomo et al., 2020; Mendler-Dünner et al., 2020; Drusvyatskiy & Xiao, 2023). Specifically, $\mathcal{D}_y(x,p)$ is assumed to be sufficiently Lipschitz in the forecast $p$. This restriction rules out common settings where decision makers choose actions that influence outcomes if the forecast $p$ is above or below some threshold. In education, for instance, counselors at a school often assign extra attention to students by examining whether their predicted probabilities are below some fixed threshold (Perdomo et al., 2023). This thresholding implies that the distribution map is not Lipschitz.

To the best of our knowledge, all subsequent work in this area also imposed some continuity restriction on $\mathcal{D}(\cdot)$ to prove that algorithms converged to stable points (e.g. (Khorsandi et al., 2024; Taori & Hashimoto, 2023)).[6] Except for the fact that $y$ is binary, we establish convergence to stability under essentially the weakest possible conditions for the outcome performative setting since $\mathcal{D}_y(x,p)$ is unrestricted.

## 5. Suboptimality of Perfectly Calibrated Performative Predictions

We end our work by showing how validity desiderata developed in supervised learning contexts (like calibration) fall short in performative settings. For a predictor to be good, it's not enough for it to forecast outcomes accurately; it needs to actively *steer* the data and make use of the fact that $f$ shapes $\mathcal{D}(f)$. In more detail, we establish the following result.

---

[6]An exception to this is the recent paper by (Wang et al., 2025). They show convergence to performative stability for a specific multi-agent performative prediction problem.

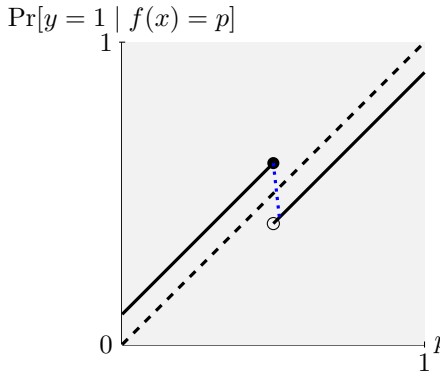

$\Pr[y = 1 \mid f(x) = p]$

*Figure 1.* The distribution map in Theorem 5.1. The solid black lines describe the probability that $y = 1$ given that the prediction is equal to $p$. The blue dotted line indicates the probability that $y = 1$ if one deploys the randomized predictor $f_r$ that mixes between $1/2$ with probability $\lambda$ and $1/2 + \varepsilon$ with probability $1 - \lambda$ for the whole range of $\lambda \in [0, 1]$. Fixed points, $\mathbb{E}_{\mathcal{D}(f)}[y] = \mathbb{E}_{\mathcal{D}(f)}[f(x)]$ are those that cross the dotted, $y = x$, diagonal line.

**Theorem 5.1.** *There exists a distribution map $\mathcal{D}(\cdot)$, such that for any $\varepsilon > 0$. There exists a randomized $f$ such that for all bounded functions $c(x, p)$ that are continuous in $p$,*

$$\left| \mathbb{E}_{(x,p,y)\sim\mathcal{D}(f)}[c(x,p)(y-p)] \right| \leq \varepsilon.$$

*Furthermore, for any function, $h : \mathcal{X} \to [0,1]$,*

$$\mathbb{E}_{(x,p,y)\sim\mathcal{D}(f)}(p - y)^2 \geq \mathbb{E}_{(x,y)\sim\mathcal{D}(h)}(y - h(x))^2 - \mathcal{O}(\varepsilon).$$

*Hence, for any class of functions $\mathcal{H} \subseteq \{\mathcal{X} \to [0,1]\}$:*

$$\mathbb{E}_{(x,p,y)\sim\mathcal{D}(f)}(p - y)^2 \geq$$
$$\max_{h\in\mathcal{H}} \mathbb{E}_{(x,y)\sim\mathcal{D}(h)}(y - h(x))^2 - \mathcal{O}(\varepsilon).$$

*Proof.* Consider the case where there are no features, and the outcome $y$ is binary. Since there are no features, we define the distribution map to be $\mathcal{D}_y(x, p) = \Pr[y = 1 \mid p] = g(p)$ where $g(p) = p + .01$ if $p \leq .5$ and $p - .01$ if $p > .5$. We depict the distribution map visually in Figure 1.

Note that there is no determistic predictor $f_p = p$ such that $\mathbb{E}_{\mathcal{D}(f_p)}[y] = \mathbb{E}_{\mathcal{D}(f_p)}[f_p]$ since $\mathbb{E}_{\mathcal{D}(f_p)}[y] = g(p)$ and there does not exist $p \in [0, 1]$ such that $g(p) = p$. However, the predictor $f_r$ that uniformly mixes between $p_1 = 1/2$ and $p_2 = 1/2 + \alpha$ for some small $\alpha$ satisfies $\mathbb{E}_{y\sim\mathcal{D}(f_r)}[y] = \mathbb{E}_{p\sim f_r}[p]$ for any $\alpha < .5$. Hence, if we don't impose continuity assumptions on $\mathcal{D}$, we need predictors to be randomized to find solutions such that $\mathbb{E}_{\mathcal{D}(f)}[y - f(x)] = 0$. Moreover, given any continuous $c : [0, 1] \to [-1, 1]$,

$$\mathbb{E}_{y\sim\mathcal{D}(f_r),p\sim f_r}[c(p)(y-p)] = \frac{.01}{2}\left(c(1/2) - c(1/2 + \alpha)\right)$$

By letting $\alpha$ go to 0, $\mathbb{E}_{(x,y)\sim\mathcal{D}(f_r)}[c(p)(y - p)] \to 0$ since $\lim_{\alpha\to 0} c(1/2 + \alpha) = c(1/2)$. This predictor $f_r$ which uniformly mixes between $1/2$ and $1/2 + \varepsilon$ also satisfies

$$\mathbb{E}_{y\sim\mathcal{D}(f),p\sim f}(y - p)^2 = 1/4 + \mathcal{O}(\alpha)$$

since $y$ is either 1 or 0 and $p$ is always nearly $1/2$. The performative risk for this problem is a quadratic function in $p$ that is maximized at $p = 1/2$. Yet the performatively optimal solutions are to predict either 0 or 1. For these, we would have the best possible performative risk of .01. $\square$

This construction complements insights from a previous result by Miller et al. (2021), who also showed how performatively stable models can maximize the performative risk. Our result provides a different perspective by connecting these notions of loss minimization (stability and optimality) with notions of computational indistinguishability. Lastly, we presented this construction where there are no features for the sake of simplicity. However, one could extend it to include features by redoing a similar conditional distribution pointwise for every $x \in \mathcal{X}$.

## 6. Discussion and Future Work

Prediction algorithms are now commonplace in important social domains and performativity abounds. By shaping our decision-making, these prediction algorithms influence the outcomes we see. Drawing on a long intellectual history, our work revisits core methodological questions regarding the design and evaluation of predictors in these domains.

On the design side, we introduced new algorithms that can overcome social feedback and produce rigorously calibrated predictions of social events. These results resolve algorithmic questions left open by (Simon, 1954) and (Grunberg & Modigliani, 1954). On the evaluation side, our contributions are mostly conceptual. We illustrate how traditional desiderata imported from supervised learning— calibration and its different variants — mean something quite different in these performative contexts. These insights call into question the utility and significance of public forecasts of social events that stake their validity on calibration (Silver, 2019).

There are a number of interesting directions for future work in this area. For one, we prove our results for the simplest case of performative prediction: (stateless) outcome performativity. It would be interesting to consider whether these results carry over to richer, stateful domains where the outcomes we see don't just depend on the predictions we make today, but also on the predictions we've made in the past (Brown et al., 2022). It would also be valuable to fully understand whether these results carry over to the setting where both feature and outcomes are performative.

## Acknowledgements

We would like to thank Aaron Roth, Tijana Zrnic, and the anonymous ICML reviewers for helpful comments and discussion. JCP was supported by the Harvard Center for Research on Computation and Society as well as by Alfred P. Sloan Foundation grant G-2020-13941

## Impact Statement

This paper presents work whose goal is to advance the field of Machine Learning. There are many potential societal consequences of our work, none which we feel must be specifically highlighted here.

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

## A. The K29 Algorithm

---



The K29 Algorithm

**Input:** A kernel $k : (\mathcal{X} \times [0,1])^2 \to \mathbb{R}$

For $t = 1, 2, \ldots$

1. Given history $\{(x_i, p_i, y_i)\}_{i=1}^{t-1}$ and current features $x_t$ define $S_t \ : \ [0,1] \to \mathbb{R}$ as

$$S_t(p) = \sum_{i=1}^{t-1} k((x_t, p), (x_i, p_i))(y_i - p_i).$$

2. If $S_t(1) \geq 0$ predict $p_t = 1$. Else if $S_t(0) \leq 0$ predict $p_t = 0$.

3. Else, run binary search to return $p_t \in [0,1]$ such that $S_t(p_t) = 0$



---

*Figure 2.* At every round, the algorithm implicitly publishes a function $f_t : \mathcal{X} \to [0,1]$ where predictions $p_t = f(x_t)$ are chosen by solving a simple optimization problem defined with respect to the history $\{(x_i, p_i, y_i)\}_{i=1}^{t-1}$. For simplicity, we state the algorithm with exact root finding. The main regret bound is still true, however, if one finds an approximate root $|S_t(p)| \leq 1/\mathrm{poly}(t)$. It degrades by an additive constant. The per round run time in $\widetilde{\mathcal{O}}(t \cdot \mathrm{time}(k))$ where $\mathrm{time}(k)$ is an upper bound on kernel evaluation.

For the sake of completeness, we provide a short overview of the K29 algorithm from (Vovk et al., 2005a;b). It is a simple, kernel-based procedure that is hyperparameter free. It guarantees online multicalibration with respect to functions $f$ in a Reproducing Kernel Hilbert Space $\mathcal{F}$. We summarize its main guarantee below:

**Proposition A.1** ((Vovk et al., 2005a)). *Let $k((x,p),(x',p'))$ be a kernel that is continuous in $p$ and let $\mathcal{F}$ be its associated RKHS $\mathcal{F}$ with norm $\|\cdot\|_\mathcal{F}$. With probability 1, the predictions $p_t$ produced by the K29 algorithm in the online protocol (Definition 3.1) result in a transcript $\{(x_t, p_t, y_t)\}_{t=1}^T$ satisfying,*

$$\Big| \sum_{t=1}^T f(x_t, p_t)(y_t - p_t) \Big| \leq \|f\|_\mathcal{F} \sqrt{\sum_{t=1}^T k((x_t, p_t)(x_t, p_t))(y_t - p_t)^2}$$

*Proof.* The proof uses basic facts about RKHS. By the reproducing property, function evaluation in the RKHS can be written as an inner product, $f(x, p) = \langle f, \Phi(x, p) \rangle_\mathcal{F}$, where $\Phi : \mathcal{X} \times [0,1] \to \mathcal{F}$ is the feature map for the RKHS and $f$ is an element in $\mathcal{F}$. Using linearity of inner products and Cauchy-Schwarz,

$$\Big| \sum_{t=1}^T f(x_t, p_t)(y_t - p_t) \Big| = \Big| \sum_{t=1}^T \langle f, \Phi(x_t, p_t) \rangle_\mathcal{F} (y_t - p_t) \Big|$$

$$= \Big| \langle f, \sum_{t=1}^T \Phi(x_t, p_t)(y_t - p_t) \rangle_\mathcal{F} \Big| \leq \|f\|_\mathcal{F} \Big\| \sum_{t=1}^T \Phi(x_t, p_t)(y_t - p_t) \Big\|_\mathcal{F}. \tag{8}$$

We now focus on bounding $\|\sum_{t=1}^T \Phi(x_t, p_t)(y_t - p_t)\|_\mathcal{F}$. By construction, the K29 algorithm always chooses a prediction $p_t$ such that:

$$\sup_{y \in [0,1]} \langle (y - p_t) \Phi(x_t, p_t), \sum_{i=1}^{t-1} \Phi(x_i, p_i)(y_i - p_i) \rangle_\mathcal{F} = \sup_{y \in [0,1]} (y - p_t) S_t(p_t) \leq 0 \tag{9}$$

Here, we used the fact that kernels represent inner products, $k((x,p),(x',p')) = \langle \Phi(x,p), \Phi(x',p') \rangle$ to do the rewriting. To see why Equation (9) holds, note that if $S_t(1) \geq 0$ then $(y-1)S_t(1) \leq 0$ for all $y \in [0,1]$. An analogous fact holds for the case where $S_t(0) \leq 0$. If neither of these is true, then $S_t(1)$ and $S_t(0)$ have opposite signs, and binary search returns a $p_t$

such that $S_t(p_t) = 0$ (which exists by continuity of the kernel). Next, we show that,

$$\| \sum_{t=1}^{T} \Phi(x_t, p_t)(y_t - p_t) \|_{\mathcal{F}}^2 \leq \sum_{t=1} \| \Phi(x_t, p_t)(y_t - p_t) \|_{\mathcal{F}}^2 = k((x_t, p_t), (x_t, p_t))(y_t - p_t)^2 \qquad (10)$$

This follows by induction on the partial sums $V_t = \| \sum_{i=1}^{t} v_i \|_{\mathcal{F}}^2$, for $v_i = \Phi(x_i, p_i)(y_i - p_i)$,

$$V_{t+1} = \| \sum_{i=1}^{t} v_i + v_{t+1} \|_{\mathcal{F}}^2 = \| \sum_{i=1}^{t} v_i \|_{\mathcal{F}}^2 + \| v_{t+1} \|_{\mathcal{F}}^2 + 2 \langle \sum_{i=1}^{t} v_i, v_{t+1} \rangle_{\mathcal{F}} \leq V_t + \| v_{t+1} \|_{\mathcal{F}}^2.$$

Here, we used Equation (9) to upper bound the cross term above. The result follows by taking square roots on either side of Equation (10) and plugging the upper bound into Equation (8). $\qquad \square$

To apply the algorithm to specific function classes $\mathcal{H}$, the only remaining todo is to show how to construct kernels $k$ with corresponding RKHS $\mathcal{F}$ such that $\mathcal{H} \subseteq \mathcal{F}$ and to show that both kernel evaluations $k((x, p), (x, p))$ and function norms $\|h\|_{\mathcal{F}}$ are uniformly bounded. This yields the $\mathcal{O}(\sqrt{T})$ regret bound. See (Dwork et al., 2025) for examples on how to do this for common function classes.

# B. Deferred Proofs

## B.1. Proof of Theorem 3.4

*Proof.* The proof follows the typical template of online to batch conversions, with the modification that we now draw samples from $\mathcal{D}(\cdot)$ rather than a static distribution $\mathcal{D}$.

Let $\{x_i, y_i, f_i, p_i\}_{i=1}^{n}$ be the sequence of random variables generated in the $n$-round interaction (online to batch conversion) where $f_i$ is chosen by the online algorithm $\mathcal{A}$ as a function of $\{x_s, y_s, f_s, p_s\}_{s=1}^{i-1}$, and the data at round $i$ is generated from sampling process $x_i \sim \mathcal{D}_x$, $p_i = f_i(x_i)$, $y_i \sim \mathcal{D}_y(x_i, p_i)$. By definition of $f_{\mathcal{A}}$ and the outcome performavity assumption on $\mathcal{D}(\cdot)$,

$$\mathbb{E}_{(x,p,y) \sim \mathcal{D}(f_{\mathcal{A}})} c(x, p)(p - y) = \sum_{i=1}^{n} \mathbb{E}_{(x, p_i, y) \sim \mathcal{D}(f_i)} [c(x, p_i)(p_i - y) \mid f_{\mathcal{A}} = f_i] \cdot \Pr[f_{\mathcal{A}} = f_i]$$

$$= \frac{1}{n} \sum_{i=1}^{n} \mathbb{E}_{\substack{x \sim \mathcal{D}_x, \\ y \sim \mathcal{D}_y(x, f_i(x))}} [c(x, f_i(x))(f_i(x) - y) \mid f_{\mathcal{A}} = f_i], \qquad (11)$$

where $f_i$ is again the predictor chosen by the online learning algorithm at round $i$ (which is measurable with respect to $\pi_{i-1} = \{(x_j, y_j, p_j, f_j)\}_{j=1}^{i-1}$) and $p_i = f_i(x)$. Now, consider the following stochastic process $(V_i)_{i=0}^{n}$,

$$V_i = V_{i-1} + \mathbb{E}_{(x, p_i, y) \sim \mathcal{D}(f_i)} [c(x, p)(p - y) \mid f_{\mathcal{A}} = f_i, \pi_{i-1}] - c(x_i, p_i)(p_i - y_i).$$

This is a martingale since $V_i$ is a function of $\pi_i$ and

$$\mathbb{E}[c(x_i, p_i)(p_i - y_i) \mid \pi_{i-1}] = \mathbb{E}_{(x, p_i, y) \sim \mathcal{D}(f_i)} [c(x, p)(p - y) \mid f_{\mathcal{A}} = f_i, \pi_{i-1}].$$

By assumption, $|c(x, p)(p - y)| \leq 1$ for any $(x, y, p)$. Hence, the increments lie in [-2,2]. Using the Azuma-Hoeffding inequality, with probability $1 - \delta$:

$$|V_n| = \Big| \sum_{i=1}^{n} \mathbb{E}_{(x,y) \sim \mathcal{D}(f_i), p \sim f_i(x)} [c(x, p)(p - y) \mid f_{\mathcal{A}} = f_i, \pi_{i-1}] - c(x_i, p_i)(p_i - y_i) \Big| \leq \sqrt{8n \log(2/\delta)}.$$

The above holds for a specific $c$ and we can have it hold for all $c \in \mathcal{C}$ via a union bound. Now using the triangle inequality and our identity from Equation (11) we get that:

$$\Big| \mathbb{E}_{(x,y) \sim \mathcal{D}(f_{\mathcal{A}}), p \sim f_{\mathcal{A}}(x)} c(x, p)(p - y) \Big| \leq \Big| \frac{1}{n} \sum_{i=1}^{n} c(x_i, p_i)(p_i - y_i) \Big| + 4 \sqrt{\frac{\log(|\mathcal{C}|) + \log(1/\delta)}{n}}$$

for all $c \in \mathcal{C}$. The result follows by upper bounding the first term on the right hand side by the regret bound on the online algorithm. $\qquad \square$

## B.2. Proof of Corollary 3.5

*Proof of a)* Consider the kernel,

$$k((x,p),(x',p')) = \sum_{c \in \mathcal{C}} c(x,p)c(x',p').$$

For any $(x,p)$, $k((x,p),(x,p)) \leq |\mathcal{C}|$ since $c(x,p) \in [-1,1]$. Furthermore, the kernel is continuous in $p$ since all the functions $c$ are assumed to be continuous in $p$. By the Moore–Aronszajn theorem, $k$ has an RKHS $\mathcal{F}$ such that $c \in \mathcal{F}$ for all $c \in \mathcal{C}$. Furthermore, $\|c\| \leq 1$. Therefore, by Proposition A.1, the K29 algorithm guarantees online multicalibration with respect to all $c \in \mathcal{C}$ with regret bounded by $\sqrt{T \cdot |\mathcal{C}|}$. Note that the kernel can be evaluated in time $\mathcal{O}(|\mathcal{C}|)$.

*Proof of b)* It is a well-known fact that the linear kernel $k((x,p),(x',p')) = \langle x,x' \rangle + pp'$ has an reproducing kernel Hilbert space $\mathcal{F}_k \subseteq \{\mathcal{X} \times [0,1] \to \mathbb{R}\}$ containing all linear functions $c(x,p) = \langle x,\theta \rangle + b \cdot p$. Moreover, the squared norm of these functions in the RKHS is equal to $\|\theta\|_2^2 + p$.

Therefore, Proposition A.1 guarantees that the K29 algorithm instantiated with this linear kernel is online multicalibrated with respect to $\mathcal{C} \subseteq \mathcal{F}_k$ at rate $\sqrt{2T}$. Since the kernel takes $\mathcal{O}(d)$ time to evaluate, the per-round runtime of the algorithm is at most $\mathcal{O}(t \cdot d)$.

*Proof of c)* The regret bound follows from the analysis in Corollary 3.3 from (Dwork et al., 2025) which provides an explicit choice of kernel such that the AnyKernel or K29 algorithms are guarantee online outcome indistinguishable with respect to all degree $s$ polynomial functions $\mathcal{C}$ at rate bounded by $10\sqrt{d^s \cdot T}$. The (ANOVA) kernel in this construction can be computed in time at most $\mathcal{O}(d \cdot s)$ (Shawe-Taylor & Cristianini, 2004), hence the bound on the runtime.

## B.3. Proof of Lemma 4.4

*Proof.* Expanding out the first term on the left-hand side,

$$\mathbb{E}_{(x,p,y) \sim \mathcal{D}(f)} \ell(p,y) = \mathbb{E}_{x \sim \mathcal{D}_x, p \sim f(x), y \sim \mathcal{D}_y(x,p)} (\ell(p,1) - \ell(p,0))y + \ell(p,0).$$

Furthermore,

$$\mathbb{E}_{\substack{x \sim \mathcal{D}_x, p \sim f(x) \\ y \sim \mathrm{Ber}(p)}} \ell(p,y) = \mathbb{E}_{x \sim \mathcal{D}_x, p \sim f(x)} [(\ell(p,1) - \ell(p,0)) \mathbb{E}_{y \sim \mathrm{Ber}(p)}[y \mid p] + \ell(p,0)]$$

$$= \mathbb{E}_{x \sim \mathcal{D}_x, p \sim f(x)} [(\ell(p,1) - \ell(p,0))p + \ell(p,0)].$$

Combining these two equations with the observation that for squared loss, $\ell(p,1) - \ell(p,0) = 1/2 - p$, we get that:

$$\mathbb{E}_{x \sim \mathcal{D}_x, p \sim f(x), y \sim \mathcal{D}_y(x,p)} \ell(p,y) - \mathbb{E}_{\substack{x \sim \mathcal{D}(f), p \sim f(x) \\ y \sim \mathrm{Ber}(p)}} \ell(p,y) = \mathbb{E}_{x \sim \mathcal{D}_x, p \sim f(x), y \sim \mathcal{D}_y(x,p)} [(\ell(p,1) - \ell(p,0))(y - p)]$$

$$= \mathbb{E}_{x \sim \mathcal{D}_x, p \sim f(x), y \sim \mathcal{D}_y(x,p)} [(-p + 1/2)(y - p)].$$

An identical argument shows that, given any $h : \mathcal{X} \to [0,1]$,

$$\mathbb{E}_{\substack{x \sim \mathcal{D}_x, p \sim f(x) \\ y \sim \mathrm{Ber}(p)}} \ell(h(x),y) - \mathbb{E}_{(x,y) \sim \mathcal{D}(f)} \ell(h(x),y) = \mathbb{E}_{x \sim \mathcal{D}_x, p \sim f(x), y \sim \mathcal{D}_y(x,p)} [(-h(x) + 1/2)(y - p)].$$

Taking absolute values, we see that these conditions are exactly equal to the requirement that $f$ is performatively multicalibrated with respect to the functions $c(x,p) = p - 1/2$ and $c_h(x,p) = h(x) - 1/2$. $\qquad \square$

