# OpenReview forum: "Revisiting the Predictability of Performative, Social Events"
_ICML.cc/2025/Conference — ICML 2025 poster_

### Official Review · Reviewer_QnUX · 2025-02-17

**Overall Recommendation:** 5

**Summary:**

The authors consider a classic problem in social science--how can we make accurate predictions about the world if our predictions affect the world--from a learning theory perspective. In this setting, there  are features $x$, a binary outcome $y$, and we wish to make probabilistic predictions $f(x)$ to predict $y$. However, once we pick $f$, the world changes so that $(x, y) \sim \mathcal D(f)$, and outcomes now depend on our predictions in an arbitrary way (the performative prediction setting). Classic work using fixed-point analysis shows that there exist predictors $f$ such that $f(x)$ predicts $y$ when $(x, y)\sim \mathcal D(f)$, but does not provide a method of computing such predictors. Here, the authors adapt online multicalibrated algorithms to performative prediction. They show that it is indeed possible to efficiently learn a predictor $f(x)$ that is (multi)calibrated (on whichever subsets of $x$ we care about) with the induced outcomes drawn from $\mathcal D(f)$. Moreover, they show a connection to outcome indistinguishability, so that their predictor's outputs are computationally indistinguishable from the world outcomes. On the other hand, through a construction, the authors also demonstrate that the learned predictor can utterly fail to explain the variance in $y$, despite being perfect calibrated. Intuitively, they show that in some cases, the only way to achieve calibration is to steer the world into higher variance outcomes, where are predictions are essentially useless (despite being calibrated). In contrast, if we were willing to accept some small bias in our predictions in this construction, we could steer the world into highly predictable outcomes (consistently y=0 or y=1).

### Update after rebuttal
I continue to think this is a very nice paper and I'm glad the other reviews agree on acceptance.

**Claims And Evidence:**

The theoretical claims are thoroughly supported by proofs, including sketches in the main text and extended proofs in the appendix.

**Essential References Not Discussed:**

None that I am aware of, although this is not my primary area of expertise. However, the paper in many places acknowledges where the ideas it draws from originated, so my impression is that the authors are very familiar with the relevant related work.

**Experimental Designs Or Analyses:**

N/A

**Methods And Evaluation Criteria:**

N/A

**Other Comments Or Suggestions:**

1. The construction from section 5 is very nice and easy to understand, and makes the distinction between performative stability and performative optimality much clearer than the definitions and Eqs (1) and (2). I would recommend describing the intuition behind this construction in 1.1, so the reader doesn't have my experience: thinking "how can this possibly be true, I don't understand how (1) and (2) lead to such drastic differences" until they reach the last page of the paper and it clicks.

**Other Strengths And Weaknesses:**

Strengths:
1. The quality of the paper and writing are extremely high.
2. The problem is a very important one, and from my understanding this paper provides a very nice contribution to the literature

Weaknesses:
1. In some cases, I found the subtle differences between equations and definitions difficult to follow, so I think there could be just a bit more intuition provided (see suggestion below for the main one)

**Questions For Authors:**

1. Some places have $(x, y)\sim \mathcal D (f)$ (e.g., line 417) while others have $y \sim \mathcal D (f)$ (e.g., line 421), and yet others have $x \sim \mathcal D (f)$ (e.g., line 381). All of those cases seem to use both $x$ and $y$, so surely they should all say $(x, y)\sim \mathcal D (f)$, right?
2. Along the same lines, some places say $p \sim f$ (line 417) and others say $p \sim f(x)$. Shouldn't it always be $p \sim f(x)$?

**Relation To Broader Scientific Literature:**

The key contribution is a method for constructing a calibrated performative predictor, and analysis of why this might not be the best measure for social prediction.

**Theoretical Claims:**

I am confident that the analysis in section 5 is correct. The other proofs sound reasonable, although I have not extensively checked the details.

---

> ### Author Rebuttal · Authors · 2025-03-31
>
> Thank you for all of the insightful comments on our work! We’re glad you found the paper interesting.
>
> We will certainly add intuition behind the construction in section 5 to the introduction. Your comments in the summary of your review are very helpful in this regard. We appreciate it!
>
> And yes, thanks for catching those typos; we will fix them.

---

### Official Review · Reviewer_Y8H3 · 2025-03-07

**Overall Recommendation:** 3

**Summary:**

This paper formalizes the question of whether predictions can remain accurate when the act of predicting affects the state of the world. The authors address this question theoretically and show that a predictor can maintain some bounded level of calibration/validity. The paper provides a bound on how far a predictor might be from accuracy and show that high quality predictions can be found in polynomial time. The paper ends by showing that while the calibration criteria which has been used through the paper has very bad worst-case performance.

**Claims And Evidence:**

I have not found errors in the claims of the paper, however, I find the paper to be quite lacking in clarity. While the introduction does an excellent job of setting out some intuition as to the problem domain I find the explanation within the paper highly devoid of both (1) intuitive explanation of each idea being used (e.g. an example of what good [multi-]calibration means in practice, contrasted with something like accuracy would help my understanding), and (2) a clear explanation of all notation: there is some assumption of understanding around certain notation standards which are then not introduced in the paper. Basic definitions, such as the distinction between p and y needs to be spelled out more clearly to firmly establish a solid grounding that a reader will use for the rest of the paper.

**Essential References Not Discussed:**

N/A

**Experimental Designs Or Analyses:**

N/A

**Methods And Evaluation Criteria:**

The theoretical framework applied to the problem seems quite fitting.

**Other Comments Or Suggestions:**

Some minor notes:

- typo in Definition 1: "to a distributions over"
- the blue line in Figure 1 is not at all friendly to the colour-blind (or any reader that prints out papers in black and white, like myself). A different line style for that segment may be worth the effort

**Other Strengths And Weaknesses:**

In general, I find the topic of this paper to be excellent and in a direction that I truly hope is developed much further. However, I find this paper to be quite inaccessible due to a lack of both thorough explanation of the basics of the paper, and a lack of examples/intuition around how to understand the concepts.

The paper suffers as a result, both in terms of clarity and any comment I can add on the significance of the paper. A section more clearly discussing the implications of your results would help me to much better understand where this paper fits in and how important it is to the literature.

**Questions For Authors:**

I rarely update my review based on a response but you are welcome to respond to any portion of my review as you wish.

**Relation To Broader Scientific Literature:**

There is discussion around the work most directly related to this topic. The paper does not include a concluding discussion section which might add some broader connection to other areas of the scientific community.

**Theoretical Claims:**

Most proofs are not in the main paper but instead included in an appendix; I did not thoroughly review them.

---

> ### Author Rebuttal · Authors · 2025-03-31
>
> Thank you for taking the time to carefully read our manuscript and provide comments. We’re delighted you find the direction interesting and look forward to more work in this area.
>
> We will happily add more clarification and intuition on the distinctions between accuracy and calibration, the notation we use, and, more generally, the main mathematical tools used in our work. The camera-ready allows a page of more space, and we will use it for this purpose. We will also add a discussion section that succinctly summarizes the implications of our work and fix these typos and color issues in the figure.

---

### Official Review · Reviewer_vAMA · 2025-03-14

**Overall Recommendation:** 3

**Summary:**

* This paper investigates multicalibration problems in performative settings.
* The models assume the performative prediction framework, where data distribution depends on the deployed model $(x,y)\\sim \\mathcal{D}(f)$.
* The main result is a convergence bound for the performative multicalibration loss, achieved through an online-to-batch reduction (Theorem 3.4). The result is modular, converting online learning algorithms with multicalibration guarantees into batch algorithms with performative multicalibration guarantees.
* Section 4 shows that predictors which are approximately performatively multicalibrated are also approximately performatively stable for the quadratic loss function. Finally, Section 5 presents a construction showing that perfectly multicalibrated classifiers can have the worst possible performance with respect to the quadratic loss.

## Update after rebuttal
Thank you for the clarifications, and I look forward to seeing the improvements in the next revision of the paper.

**Claims And Evidence:**

Claims seem to be supported by theoretical evidence.

**Essential References Not Discussed:**

Key results in this area seem to be discussed.

**Experimental Designs Or Analyses:**

The paper does not include any empirical experiments. Although this is common in theory-focused work, a simulated example or a practically motivated case study could help demonstrate the utility of the proposed framework.

**Methods And Evaluation Criteria:**

The paper does not contain an empirical evaluation.

**Other Comments Or Suggestions:**

* Including a practical example or simulation would help illustrate the value of the proposed method compared to existing work, and build intuition about its theoretical and practical behavior.
* In addition, I wonder if the paper can benefit from an empirical demonstration of the results.

**Other Strengths And Weaknesses:**

Additional strenghts:
* The paper offers a novel convergence bound for performative multicalibration loss, providing a modular framework to bridge online and batch settings.

Additional weaknesses:
* The exposition is challenging to follow in parts, which may hinder accessibility.
* The absence of a practically motivated example or empirical demonstration limits the intuition behind the theoretical results, and the practical applicability of the findings.

**Questions For Authors:**

* Would the theoretical results still hold in a stateful performative prediction setting, such as the one considered in Brown et al.'s “Performative Prediction in a Stateful World” (AISTATS 2022)?
* Can you provide an example or simulation that demonstrates the practical advantages of your approach over traditional methods under the performative prediction framework?

**Relation To Broader Scientific Literature:**

The paper positions itself against classic results in the performative prediction literature, which typically rely on strong Lipschitz conditions. Here, the authors show that performative multicalibration is achievable under milder assumptions on the distribution map $\\mathcal{D}$.

**Theoretical Claims:**

The proofs were checked at a high level. While the high-level structure appears sound, a meticulous examination can help provide additional verification.

---

> ### Author Rebuttal · Authors · 2025-03-31
>
> We appreciate you taking the time to read and carefully critique our work. These are great questions.
>
> Stateful world. We had not considered this possibility. We don’t believe it applies directly to the stateful case, but this is an interesting, open question for future work.
>
> Also, given the additional space that comes with the revision, we’d happily include further discussion and comments to explain the main ideas behind the results and make the paper more accessible. We will also consider potential comparisons to previous algorithms and practical examples to illustrate how our ideas compare to previous approaches, with the caveat that previous approaches lack theoretical guarantees in our setting.

---

### Official Review · Reviewer_kMPD · 2025-03-17

**Overall Recommendation:** 4

**Summary:**

The paper at hand claims to explore the predictability of socil events. Social predictions do not merely describe the future. They also influence it. Such predictions can affect market prices, voter behavior, and policy outcomes. This interaction complicates the ability to forecast accurately. Early theorists, including Morgenstern and Simon, discussed these challenges extensively. This paper addresses these questions anew, using recent concepts from machine learning. Specifically, it utilizes the framework of "performative prediction," where predictions themselves shape outcomes. The authors establish that accurate forecasting of binary social events remains computationally feasible despite these dynamics. A key contribution is demonstrating the existence of predictors that remain accurate even as they actively shape events. These predictors satisfy strong conditions like multicalibration and outcome indistinguishability. Algorithms to achieve these conditions are presented, which are efficient both statistically and computationally.

However, the paper also identifies critical limitations. Although accurate predictions are always achievable, they might not always lead to desirable outcomes. Calibrated predictions may sometimes create poor social equilibria. Such predictions could, paradoxically, maximize prediction error measured in terms of performative risk. Thus, the paper shows a tension between accuracy and social desirability in forecasting social events. The authors conclude by suggesting a reconsideration of historical forecasting methods. They call for further research into what goals predictions in social contexts should ultimately serve.

**Claims And Evidence:**

All technical claims are proven. My only concern is with the general, conceptual claim of the paper as stated in the title "Revisiting the Predictability of Social Events". I understand the authors want to relate to the famous article by Grunberg and Modigliani, but I think the title does oversell the paper a little bit. The paper does not answer the questions whether social events are predictable *per se*. The paper answers the question whether social events can be predictable if these predictions have performative effects on the population. The framework is the very specific (although without the strong Lipschitz-condition on the distributions mapping) original performative prediction setup (without e.g. more realisitic stateful world as in https://proceedings.mlr.press/v151/brown22a/brown22a.pdf)

**Essential References Not Discussed:**

Discussion of https://proceedings.mlr.press/v151/brown22a/brown22a.pdf would be nice, since I consider this extension a much more realistic model of reality than the original performative prediction setup, see question below. But it is not strictly essential.

**Experimental Designs Or Analyses:**

no experiments, see above.

**Methods And Evaluation Criteria:**

This is a theoretical paper, so no experiments are needed.

**Other Comments Or Suggestions:**

typos:
"Converstion" → "Conversion"
"instatiating" → "instantiating"
"indistiguishability" → "indistinguishability"
 "ℓ(p,y)=½(1−p)²" should be "ℓ(p,y)=½(y−p)²"

**Other Strengths And Weaknesses:**

none

**Questions For Authors:**

1. conceptual question: I understand the results presented in this paper require much weaker conditions (e.g. no Lipschitzness of distribut. mapping, which in turn induces e.g. strong convexity of loss if I remember correctly?) than in the original performative prediction setup (2020 paper). So that's certainly a valid and substantial contributions of its own. In terms of conceptual results, however, I must admit that I do not find it surprising for a classifier to be multi-calibrated in a setup where it was shown before that it converges to a classifier that is close (in parameter space) to the performatively optimal one (thm. 4.3. in https://arxiv.org/pdf/2002.06673). Can the authors comment in more details on this relation?

2. Can the authors provide some intuitions why the results require *randomized* predictors, while readers are "encouraged to think of them as deterministic"?

3. For the main results (Thm. 3.4) the authors write: "The main difference here is that samples (xt, yt) are not drawn i.i.d from a fixed distribution D, but rather from the distribution (xt, yt) ∼ D(ft) induced by the predictions. Despite these differences, a similar strategy suffices." The authors then procede by presenting the results. Can they explain WHY this strategy suffices? Generally, I think the paper could benefit from some more motivation/explanation

4. Does Thm. 3.4 also hold for performative prediction in a stateful world (see reference above, i.e., in a setup where the distributions map is bivariate mapping from both predictions AND previous distributions)?

**Relation To Broader Scientific Literature:**

see above

**Theoretical Claims:**

I did spend only 1-2 hours checking the proofs, but I did not find any errors.

---

> ### Author Rebuttal · Authors · 2025-03-31
>
> Thank you for taking the time to read and provide detailed feedback on our manuscript. It is very appreciated.
>
> Re: Brown et al. 2022. This is an excellent paper. We mistakenly did not include it but will happily discuss it in the revision.
>
> Re: Conceptual question. Thanks for raising this. Lipschitzness of the distribution map D() and strong convexity of the loss are distinct conditions. One does not imply the other. Also, as per the example in Miller et al ICML 2021, the bound from Thm 4.3 in Perdomo et al. can be vacuous in the sense that the other parameters (Lipschitz constant of the loss) involved make it so the bound is just the diameter of the space. Hence, even if the distribution map is lipschitz, it does not imply that stable and optimal classifiers are close in parameter space. Of course, none of these arguments apply in our setting since we make no regularity assumptions on D(). We thank you for raising it and will add further discussion on this point in the updated version.
>
> Re: Randomized predictions. Randomization is, in general, necessary to guarantee performative calibration as per the example in Section 5. In particular, no deterministic prediction p in [0,1] has the property that E_{p}[Y] = p. One needs to randomize between different forecasts to achieve on average calibration. Our comment about near determinism was regarding the per-time-step predictions on the online algorithms. We see how this can be confusing and will clarify it. Thanks for bringing it up.
>
> Re: Explanation. We will happily clarify what we meant regarding how martingale arguments developed in online to batch conversions for supervised learning settings are close to those we develop for the performative case.
>
> Re: Stateful world. This is a great question that we had not considered. We don’t believe our results apply directly to the stateful case. However, it is an interesting question for future work.

---

> > ### Comment · Reviewer_kMPD · 2025-04-01
> >
> > Thanks for your replies. Greatly appreciated! In particular, thanks for clarifying the relation between Lipschitz condition on D() and strong convexity condition on the loss in the initial perf. pred. paper. I got that wrong initially.
> >
> > I agree that the stateful world setup might be interesting for future work, but is beyond the scope of this paper.
> >
> > All in all, I think it would be almost insane not to accept this paper.

---

### Decision · Program_Chairs · 2025-05-01

**Decision:**

Accept (poster)

**Comment:**

This paper studies the predictability of social event within a modern performative prediction setting. The paper theoretically establishes positive results for the binary case within this setting, while highlighting its limitations. Authors were generally positive about the paper, were appreciative of the aim of recreating classic results in contemporary settings, and found the theoretical results to be sound and interesting. Some reviewers encourage the authors to better frame their contribution in the precise setting in which results hold, namely stateful performative prediction, rather than a general thesis about the predictability of social events in general. Statefulness in particular seems to be a common simplifying assumption in the performative prediction literature, but that does not necessary apply broadly. Several reviewers found clarity to be improvable on several fronts – the authors are encourage to take these into account in their next revision. Adding a final discussion section is also advisable as it is the norm for almost all ICML papers.